# Deformation twinning and grain partitioning in a hexagonal close-packed magnesium alloy

M. Arul Kumar[1], B. Clausen [1], L. Capolungo[1], R.J. McCabe[1], W. Liu[2], J.Z. Tischler[2] & C.N. Tomé[1]

Pervasive deformation twinning in magnesium greatly affects its strength and formability. The local stress fields associated with twinning play a key role on deformation behavior and fracture but are extremely difficult to characterize experimentally. In this study, we perform synchrotron experiments with differential-aperture X-ray microscopy to measure the 3D stress fields in the vicinity of a twin with a spatial resolution of 0.5 micrometer. The measured local stress field aids to identify the sequence of events involved with twinning. We find that the selected grain deforms elastically before twinning, and the twin formation splits the grain into two non-interacting domains. Under further straining one domain of the grain continued to deform elastically, whereas the other domain deforms plastically by prismatic slip. This heterogeneous deformation behavior may be mediated by the surrounding medium and it is likely to lead to asymmetric twin growth.

[1] Materials Science and Technology Division, Los Alamos National Laboratory, Los Alamos, NM 87545, USA. [2] Argonne Photon Source, Argonne National Laboratory, Argonne, IL 60439, USA. Correspondence and requests for materials should be addressed to M.A.K. (email: marulkr@lanl.gov)

Hexagonal close packed (HCP) materials in general, and Mg and its alloys in particular, undergo deformation twinning in addition to slip when strained[1–3]. Deformation twins start as embryos at grain boundaries, where stress concentrations and source defects are predominately located[4–8]. Under continued straining, twins expand in forward, normal and lateral directions to reach a length scale comparable to the grain size. The forward expansion (propagation) in the in-twin-plane directions drives the twin to the opposite grain boundaries that are intersecting the twin plane and splits the grain. This propagation, however, takes place with simultaneous normal growth (thickening) and lateral growth. Normal growth displaces the coherent twin interface via a disconnection-mediated process, and lateral growth expands the lateral interface until it is arrested at other grain boundaries. This process is similar to the generalized 3-D propagation of a dislocation loop but differs in that it creates a new domain in the grain and splits the grain in two. The obvious effect of twinning is that the new twin domain usually adopts a hard-crystallographic orientation with respect to the applied deformation affecting strain-hardening behavior significantly[9–13]. And also, the formation of twin interfaces creates a barrier for the propagation of dislocations or other twins, leading to a Hall-Petch-type grain size effect on further deformation behavior[14–17]. In addition to that, interactions of other defects with twin boundaries increase the likelihood for void nucleation, cracking, and premature failure[18–20].

Understanding the above-mentioned twinning effects on material behavior demands to study of the local processes that take place in individual grains where twinning occurs. Past research concentrated mostly in characterizing these processes with transmission electron microscopy (TEM) on post-deformation specimens[21–24]. Most recently, in situ experiments[7,25–29], full-field crystal plasticity modeling[30–35] and atomistic simulations[5,36,39] have expanded the understanding of twin formation and its effect on material behavior. There is an interrelation between twin domain, twin interface, crystallographic defects and local stress fields that conditions twin propagation and hardening. A twin transformation (~13% shear for $\{10\bar{1}2\}$ tensile twin in Mg) induces local stress fields in the vicinity of twins[35–40]. The magnitude of these local stresses is determined, in part, by the ability of neighboring grains to accommodate the transformation. Recently, experimental and modeling efforts were devoted to characterize the twin local stresses and the role of neighbors[7,29,32,35,40–47]. Among all the experimental literature, only the work of Balogh et al.[44] focuses on the local elastic strain within and in the vicinity of a twin inside a bulk polycrystalline material. But they only measured one component of elastic strain, along an arbitrarily chosen diffraction vector. While this procedure shows the presence of a gradient in the vicinity of the twin, it does not allow one to derive the full elastic strain tensor and the associated stress tensor. Knowledge of the latter is essential to extract meaningful conclusions from the experiment and to assess the effect of local stress distribution on further twin thickening. In addition, in situ optical microscopy evidence shows that twin lamellae tend to grow asymmetrically, with one side propagating significantly more than the other side[27,48,49]. However, it is not known whether this asymmetry is due to differences in residual defect content on either of the twin interfaces, to substantial differences in the stress state on either side of the twin, or to both. Answering the question of how the presence of twins alters local deformation processes in the grain is relevant because the answer may challenge the current understanding that the twin simply divides the grain into separate domains of the same orientation that deform in similar manner.

In the present work, we present experimental evidence that twinning significantly changes the deformation behavior of the parent grain. We employ spatially resolved in situ synchrotron X-ray micro-diffraction using differential-aperture X-ray microscopy to quantify the 3D stress field in the vicinity of a twin in Mg alloy AZ31. The measurements are performed under fixed applied load. The experimental results yield all of the components of the stress tensor in the vicinity of the twin and, to our surprise, indicate that the grain deforms quite heterogeneously. Specifically, we find that the twin formation splits the grain into two non-interacting domains one of which, under further deformation, continues to deform elastically while the other undergoes plastic deformation by prismatic slip. The variation in the micromechanical fields on either side of the twin may lead to asymmetric twin growth.

## Results

**Twin microstructure**. To understand the local processes associated with twinning, we need to measure the spatially resolved local stress fields in the vicinity of a twin for a grain at the interior of a bulk sample. To accomplish that we have employed the differential-aperture scanning X-ray microscopy (DAXM) technique[50–52]. This technique has been developed at the 34-ID-E beamline of the Advanced Photon Source (APS), Argonne National Laboratory. In this method, X-rays penetrate several hundreds of micrometers inside a bulk magnesium sample and diffract from an interior volume. Penetration depends on the material properties and photon energy. The differential-aperture scanning (DAS) technique provides the spatial resolution along the line of illumination. In DAS, a thin platinum wire, which is a highly efficient X-ray absorber, is placed above the sample to mask a region of the diffracted beam. By translating the platinum wire with sub-micrometer step size, we collect a series of diffraction images. Reconstruction of diffraction images by subtracting the diffracted intensity at each detector pixels using the successive DAS images, enables the generation of a diffraction maps along a line inside the bulk sample with a resolution of 0.5 μm. Note that the X-ray beam is a square beam of 0.5μm size. Repeating the process for successive parallel lines produces diffraction maps of a region inside the bulk sample[50–52].

In this study, we use magnesium alloy AZ31 with a strong initial basal texture and average grain size of ~14 μm. For diffraction experiments, a $6.0 \times 3.1 \times 2.6$ mm $(RD \times TD \times ND)$ rectangular sample was cut from the rolled plate and compressed along the rolling-direction favoring activation of easy $\{10\bar{1}2\}$ tensile twinning via Poisson's extension of the main basal texture component along the ND. The sample geometry is shown in Fig. 1a. The X-ray beam is directed at 45 degrees onto the TD-RD plane with a projection along the TD, and thus is parallel to the depth coordinate of the diffraction map. The platinum wire is positioned parallel to the RD, which coincides with the width coordinate of the diffraction map, and it translates along the TD closely above the sample surface during the depth scanning. For this combination of X-ray and Pt wire position, the mapping area is always at 45 degrees to the sample surface as shown in Fig. 1a. The reference for the depth coordinate is depth = 0 at the sample surface with an angle of 45 degree, and so the depth values are a factor of √2 larger than the actual depth below the surface. More details about the experimental setup can be found in Balogh et al.[44] and Liu et al.[52] Refer to Larson et al.[50] and Levine et al.[51] for the DAXM technique used in this work.

With the sample under compression along the RD, a polychromatic X-ray beam is used to identify a region to be subjected to local strain mapping. The reconstructed images of depth resolved voxels provide Laue patterns yielding the crystallographic orientation at each point. A crystal orientation image of Mg alloy AZ31 is shown in Fig. 1b along with a pole figure

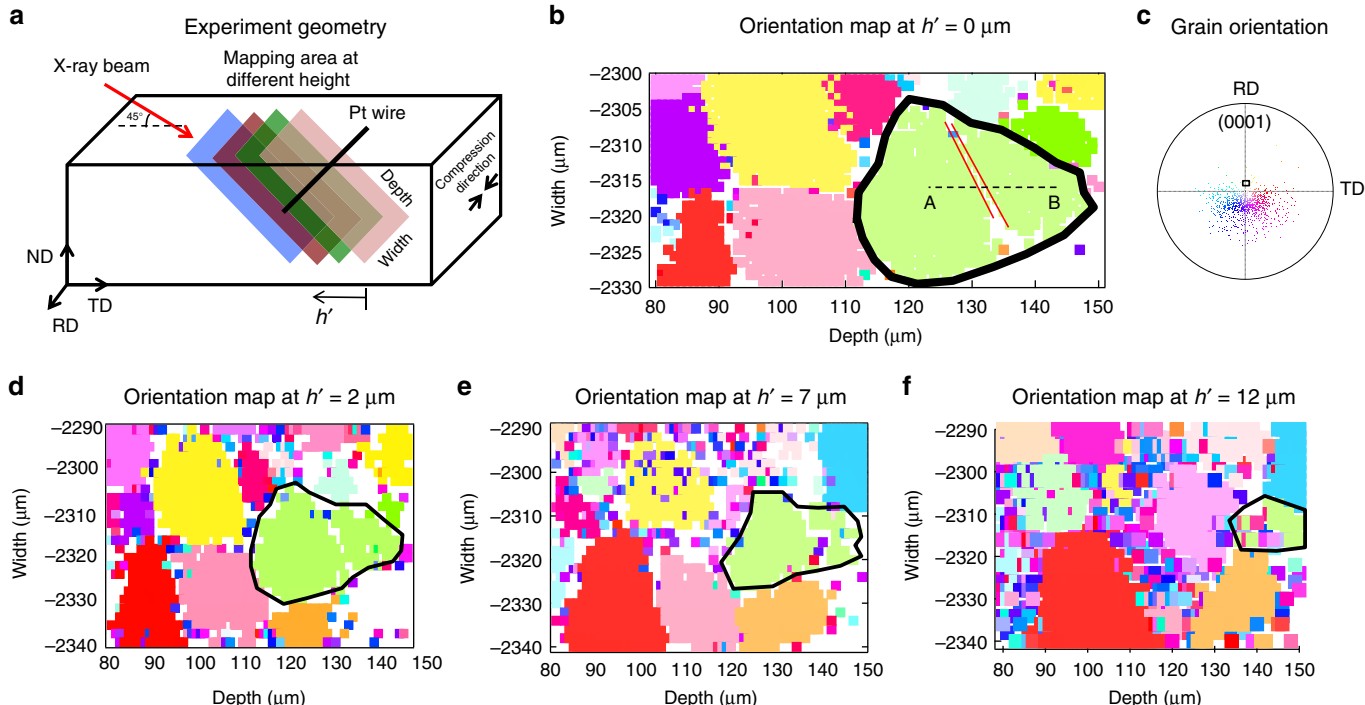

**Fig 1** Orientation mapping of twin microstructure using polychromatic beam. **a** Schematic representation of the experimental geometry and mapping area at different height. **b** Orientation map of the region of interest with 1 μm step size. The selected grain with twin is highlighted. **c** Basal pole figure of the orientation map for all the voxels present in the scan. The orientation of the selected grain for strain measurement is marked in the pole figure. **d**–**f** Orientation maps of the same region of interest at different height with 2 μm step size

representing the grain orientations in Fig. 1c. The orientation map (OM) is collected and reconstructed over a region of $250 \times 60$ micrometers using a step size of 1 micrometer. The microscopy image shown in Fig. 1b is a subset of the original scan showing the grain selected for strain measurement. The pole figure shown in Fig. 1c includes orientations of all grains present in the scan and the selected grain orientation is marked explicitly. The grain size distribution is somewhat heterogeneous and there is a strong basal texture along the ND. The color scheme for every voxel represents a crystal orientation with respect to the pole figure shown in Fig. 1c. The white regions are the voxels that are not indexed to a crystallographic orientation in the Laue pattern. The orientation of the selected grain highlighted in Fig. 1b is (0.54°, 0.97°, −9.94°) in Bunge convention (c-axis of the grain, i.e., [0001] direction nearly parallel to ND), and it has a stable {10$\bar{1}$2} tensile twin of thickness ~1 micrometer, indicated with the solid red lines in Fig. 1b). To visualize the 3D microstructure of the grain and twin, we have acquired similar OMs with a step size of 2 micrometers at different heights, $h$, shown in Fig. 1d–f.

**Local stresses in the vicinity of twin**. In order to determine the strain associated with a sub-micrometer domain, one cannot use the more conventional transmission technique based on a polychromatic beam and analyze the distortion of the Laue pattern[25,28,53]. Thus, DAXM in monochromatic mode is employed to measure the strain along a set of diffraction vectors directly from the selected grain. This method requires at least three different diffraction vectors to reconstruct the full strain tensor. The diffraction vectors for the selected grain are chosen from the polychromatic Laue pattern. The procedure for calculating the distorted unit cell is described in the Methods section. At the 34-ID-E facility three separate detectors are used, one positioned at the top and the other two tilted at the sides, eventually capturing diffracted intensity over a large region[52].

In this experiment, we have selected the following diffraction vectors for monochromatic diffraction, one from each of the three detectors: (1,1,−8), (−2,2,−9) and (2,−2,−9). We rely on Bragg's law for tuning the wavelength of the monochromatic beam to the energy of the selected diffraction vectors (17.15 keV, 14.99 keV and 14.45 keV, respectively). Monochromatic mode differential aperture scanning is performed along a line A-B, shown with dashed black lines in Fig. 1b, for all three diffraction vectors using a step size of 0.5 micrometer. The line A-B is chosen such that it is inside the grain and crosses the twin lamella, with minimal effects from neighboring grains. In what concerns the twin lamella, the diffraction data has a low intensity to noise ratio and it is omitted from the analysis. The twin thickness along line A-B is ~0.5 micrometer and it leads to the omission of one data point in the twin region. As for the points adjacent to the twin lamella, we have checked carefully that they have sufficient diffraction counts for line profile analysis.

Reconstruction of monochromatic diffraction images using the DAXM technique and 3D Gaussian fitting of line profiles in reciprocal space provides spatially resolved unit cell lattice vectors in reciprocal space. These are then converted into direct space[54]. On the basis of this, the unit cell parameters $a_1$, $b_1$, $c_1$, $\alpha_1$, $\beta_1$ and $\gamma_1$ of the locally distorted HCP crystal are calculated at every material point. Here $a_1$, $b_1$ and $c_1$ are the magnitude of the HCP unit cell vectors, and $\alpha_1$, $\beta_1$ and $\gamma_1$ are the angles between HCP unit cell vectors. Using the stress-free crystal unit cell parameters ($a_0 = 0.31985$ nm, $b_0 = 0.31985$ nm, $c_0 = 0.51957$ nm, $\alpha_0 = 90°$, $\beta_0 = 90°$ and $\gamma_0 = 120°$) for Mg alloy AZ31, and the computed parameters of the elastically deformed unit cell, the spatially resolved full elastic strain tensors are calculated[55]. Here the stress-free crystal unit cell parameters are obtained from the same sample before loading at the same beam line. The full stress tensor is then calculated using Hooke's law: $\sigma_{ij} = C_{ijkl}\, \varepsilon_{kl}$, where $C_{ijkl}$ is the stiffness tensor. The elastic constants of Mg in Voigt

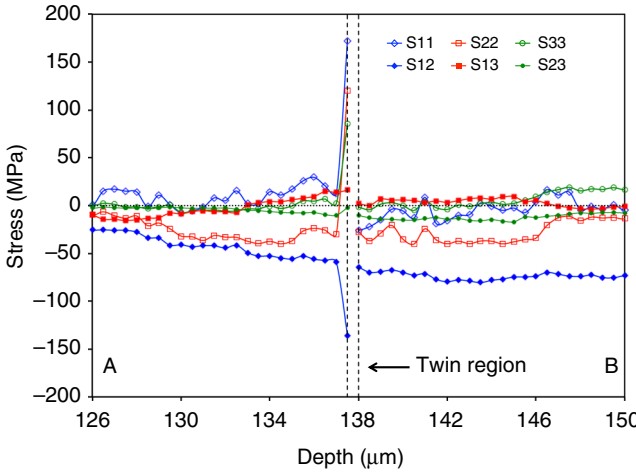

**Fig 2** Distribution of stress and strain components. Spatial distribution of elastic strain and stress components along a line A-B, shown in this figure that crosses tensile twin. The directions 1, 2 and 3 correspond to [11$\bar{2}$0], [01$\bar{1}$0] and [0001] directions of crystal coordinate system, respectively. The step size is 0.5 µm. Strong stress and strain localization observed very close to twin. The error bar for all stress components are calculated using Monte-Carlo error propagation analysis and is within ± 3.0 MPa

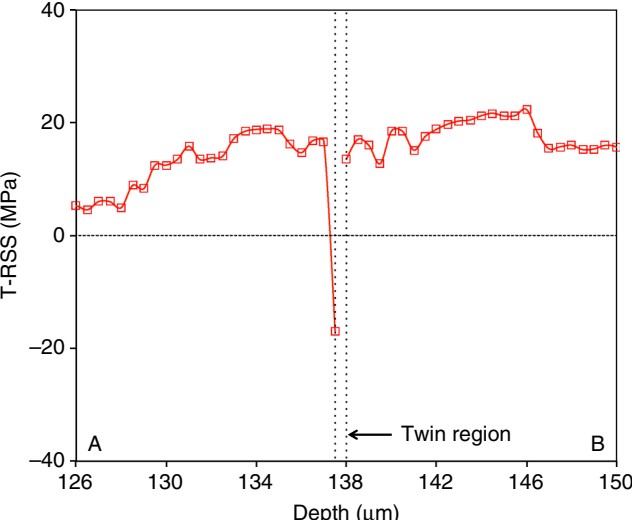

**Fig 3** Twin-plane resolved shear stress distribution. Distribution of twin plane resolved shear stress with the step size of 0.5 µm across tensile twin. The material point next to twin boundary on the left side only shows the stress reversal, and it may associated with twinning shear accommodation. And so, the effective plastic zone size for twin may be <0.5 µm

notation are (in GPa): $C_{11} = 59.75$; $C_{12} = 23.24$; $C_{13} = 21.70$; $C_{33} = 61.70$; and $C_{44} = 16.39$[56].

The calculated stress components in the crystal coordinate system (1-2-3 corresponds to [11$\bar{2}$0]−[01$\bar{1}$0]−[0001]) along the line A-B are shown in Fig. 2. The uncertainties associated with measurements and post-processes have been estimated using Monte-Carlo error propagation analysis to be within ± 3 MPa, which is about the size of the symbols, and thus they are not shown in the figure. Note that the compression direction is at an angle of ~9 degrees from the [01$\bar{1}$0] axis (direction 2) of crystal coordinate system. The measured stress profile in direction-2 for the selected grain, S22, is compressive, and thus confirms that the selected grain is well oriented for {10$\bar{1}$2}twinning. The average stress components at each side of the twin are different and, in addition, they show different spatial variation. However, the stress values adjacent to the twin lamella on the left side are significantly different than the other points in the domain, which may be a consequence of the local processes associated with twinning. The deviation of these local stress components from one part of the grain to other highlights the effects of neighboring grain interactions. Considering that Mg is nearly elastically isotropic, it is only the anisotropic plasticity that needs to be accommodated across boundaries, and induces the dispersion in stress.

**Twin induced stress-reversal**. To investigate the potential for further twin expansion, we calculate the twin resolved shear stress (T-RSS) on the plane of the active twin system. The T-RSS profile along line A-B is shown in Fig. 3. The T-RSS is positive in most of the region except at a point next to the left side of the twin lamella. The high negative value of T-RSS next to the twin interface is attributed to the stress-reversal induced by the twinning shear transformation[40]. Our recent elasto-visco-plastic Fast Fourier Transform calculations indicate that the amount of stress reversal at the twin interface for Mg alloy AZ31 is in the order of ~30 MPa[57], which is close to what we measured in this experiment. The size of the zone affected by the stress-reversal seems to be within the resolution of our measurements, that is, 0.5 to 1.0 micrometer. Further detailed experiments close to the twin

interface with a finer step size would be required to confirm the size of the twin affected zone.

**Grain partitioning**. To investigate plasticity in the vicinity of the twin, we have calculated the resolved shear stress in the basal <a>, the prismatic <a>, and the pyramidal <c+a> slip systems in the parent. The RSS profile along line A-B for individual slip systems is shown in Fig. 4. In these figures, the reference line represents the critical resolved shear stress (CRSS) needed to activate the slip system. These reference stress values correspond to stress-free initial conditions. The reference CRSS values $\tau_0$ for basal, prismatic and pyramidal follow from Dislocation Dynamic simulations[58], and are 20, 65 and 105 MPa, respectively. In Fig. 4, two lines corresponding to ±10% of the reference value are included, to account for variations in the microstructure and reference stress. The RSS profile of the basal slip systems is heterogeneous and below the threshold stress band, except for possibly a few points on the right-hand side of the twin lamella. On the other hand, the prismatic slip systems show a systematic variation: on the left side of the twin, the RSS values are smaller than the threshold values, and on the right side of the twin, the RSS of one prismatic slip system is always greater than the threshold value. The RSS profile for the pyramidal slip mode is very similar to that of prismatic but the values are much smaller than the threshold for this mode. In summary, the parent to the right of the twin likely exhibits plastic deformation by prismatic slip, and the parent to the left of the twin only deforms elastically. The activation of only prismatic slip on one side of the grain is also confirmed by the lattice rotation. We estimate the latter by comparing the orientation of the voxels on the left and right side of the twin. The measured lattice rotation values are: W12 = 0.0345; W13 = 0.0005; and W23 = 0.0022, which corresponds to a ~2.0 degree rotation of the crystal about [0.0633; −0.0017; 0.9980] direction. This particular lattice rotation can be achieved only by the activation of (01$\bar{1}$0)[2$\bar{1}$$\bar{1}$0]prismatic slip. In addition, the parent material point next to the twin on the left side is predicted to have undergone severe deformation by prismatic slip and local hardening, most likely associated with relaxation of the local stress at the twin-grain interface. In a nutshell, the twin domain splits the grain into two subdomains with uncorrelated

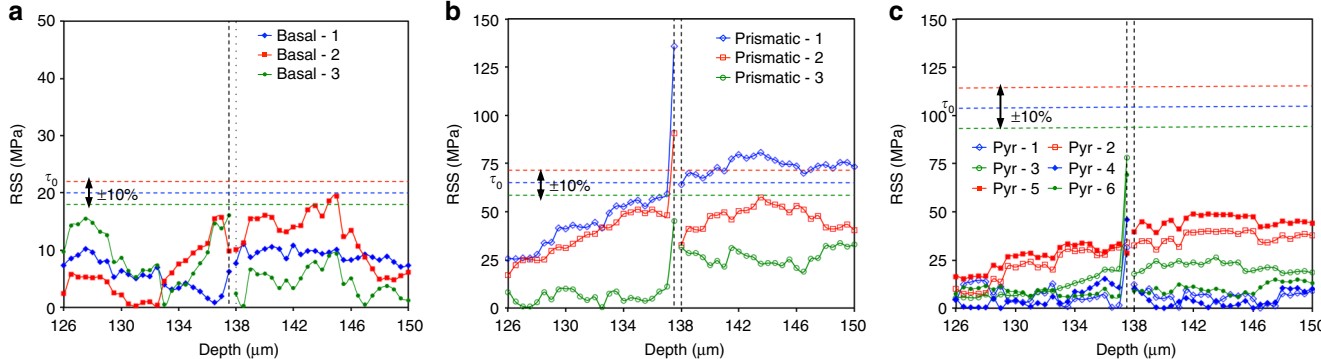

**Fig 4** Asymmetric slip activity. Spatial distribution of resolved shear stress for (**a**) basal <a>, (**b**) prismatic <a>, and (**c**) pyramidal <c+a> slip modes. The critical resolved shear stress (CRSS) for each mode is shown with a 10% band to identify the activation of slip modes. Left side of the twin seems to completely elastic and on the right side only prismatic slip is possible to activate

deformation modes. The deformation of these two domains is, no doubt, influenced by the surrounding grain neighborhood. The heterogeneous deformation behavior of the twinned grain likely has important implications on mechanical behavior and further twinning. Particularly, the asymmetry in the plastic slip activity and in the dislocation population between left and right part of the grain may lead to asymmetric twin thickening under further straining.

In addition, understanding the role of the sequence of activation of plastic deformation modes, including twinning, is relevant for developing and/or to improving modeling capabilities. Our study suggests that before twinning, we anticipate that the entire grain may have deformed relatively homogeneously except near grain boundaries (due to inter-grain interactions). From the measurement, we observe that the grain domain on the left side of the twin continues to deform elastically even after twinning. And so, before twinning the entire grain may deform elastically. Local dislocation reactions at the grain boundaries, or slip or twin activity in the neighboring grain may lead to twin nucleation in the selected grain. The material points in the vicinity of the twin tip in the neighboring grain are not indexed successfully at either end, which suggests that these regions are more heavily deformed. The twin in the selected grain may induce concentrated plasticity in the neighboring grains or, alternatively, slip/twin localization in the neighboring grain may trigger twinning in the selected grain. After twinning the grain domain on the left side deforms elastically and the one on the right side deforms plastically. This experiment nicely provides a basis to establish the sequence of deformation events taking place in the twinned grain in the bulk, which is not possible to get from other techniques. Note that this can be done using electron microscopy only for thin samples via controlled complicated in situ experiments. Current modeling efforts typically do not address the importance of the competing kinetics of each dissipative process. In order to do so, computational models would to correctly capture the nucleation and propagation of the twin, the associated stress field relaxation, and interaction with neighboring grains. Thus, the present experimental work helps to understand the complicated local deformation mechanisms and to guide the modeling tools.

**Dislocation densities and prismatic sensitive plasticity.** To further understand the plasticity in the twinned grain, we have calculated local dislocation densities associated with individual slip systems. The link between resolved shear stress and dislocation density is given by the generalized Taylor law originally proposed by Franciosi and Zaoui[59] and validated for Mg by

Bertin et al.[58] using Dislocation Dynamics. The law is:

$$\tau^s = \tau_0^s + \mu^s b^s \sqrt{\sum_{s'} h_{ss'} \rho^{s'}}, \tag{1}$$

where $\tau_0^s$, $\mu^s$, $b^s$ and $\rho^s$ are the threshold shear stress, shear modulus, magnitude of Burgers vector, and dislocation density in slip system s, respectively. The reference dislocation density $\rho^{ref}$ is assumed to be $10^{12}$ m$^{-2}$ and the shear modulus for Mg is 16.39 GPa. The value of $b^s$ is 3.1985 Å for <a> slip and 6.077 Å for <c+a> slip. And $h_{ss'}$ are the interaction coefficients between slip system s and s'. The values of $h_{ss'}$ for the different dislocation configurations in Mg can be found in Table 11 in Bertin et al.[58] For the threshold stress value $\tau_0^s$ we assume a range of ±10% of the CRSS values listed before (Fig. 4), to account for dispersion induced by the intragranular microstructure. The resolved shear stress for each of the slip systems follows from processing the diffraction data and is shown in Fig. 4. For the range of threshold stress values, $\tau_0^s$, the dislocation density is calculated from the measured resolved shear stress using Eq. (1). We perform this calculation for all slip systems belonging to the basal <a>, prismatic <a> and pyramidal <c+a> slip modes. However, we calculate the dislocation density only for those systems where the resolved shear stress, $\tau^s$, is greater than the threshold shear stress value, $\tau_0^s$. For the other slip systems, we assume the reference value ($\rho^s = 10^{-12}$ m$^{-2}$). As already mentioned, we find that the $(01\bar{1}0)[2\bar{1}\bar{1}0]$ prismatic slip system is the only active system in the region along line A–B. The normalized dislocation density ($\rho^s/\rho^{ref}$) of that prismatic slip system is shown in Fig. 5. The dislocation density ratio along line A–B is shown in Fig. 5a as a function of the threshold shear stress values. As noted earlier, the left side of the twin deforms elastically and so its dislocation density ratio is one. On the right side, the dislocation density ratio varies between 1.0 and $10^3$, indicating that the dislocation density of prismatic slip varies from $10^{12}$ to $10^{15}$. The dislocation density varies significantly depending on the reference value chosen for prismatic slip, but very little for basal slip (see Fig. 5b, c). In Fig. 5b, the dislocation density ratio for fixed basal slip threshold stress value (=actual CRSS = 20 MPa) is shown for a range of prismatic slip threshold stress values. On the other hand, in Fig. 5c, the dislocation density ratio for fixed prismatic slip threshold stress value (=actual CRSS = 65 MPa) is shown for a range of basal slip threshold stress values. Figure 5b, c corresponds to sections shown schematically in Fig. 5a. The dislocation density ratio distribution in Fig. 5b, c clearly reveals that the dislocation density varies significantly with prismatic slip mode strength, but not so with basal slip mode strength. To further understand the influence of prismatic slip mode strength, we

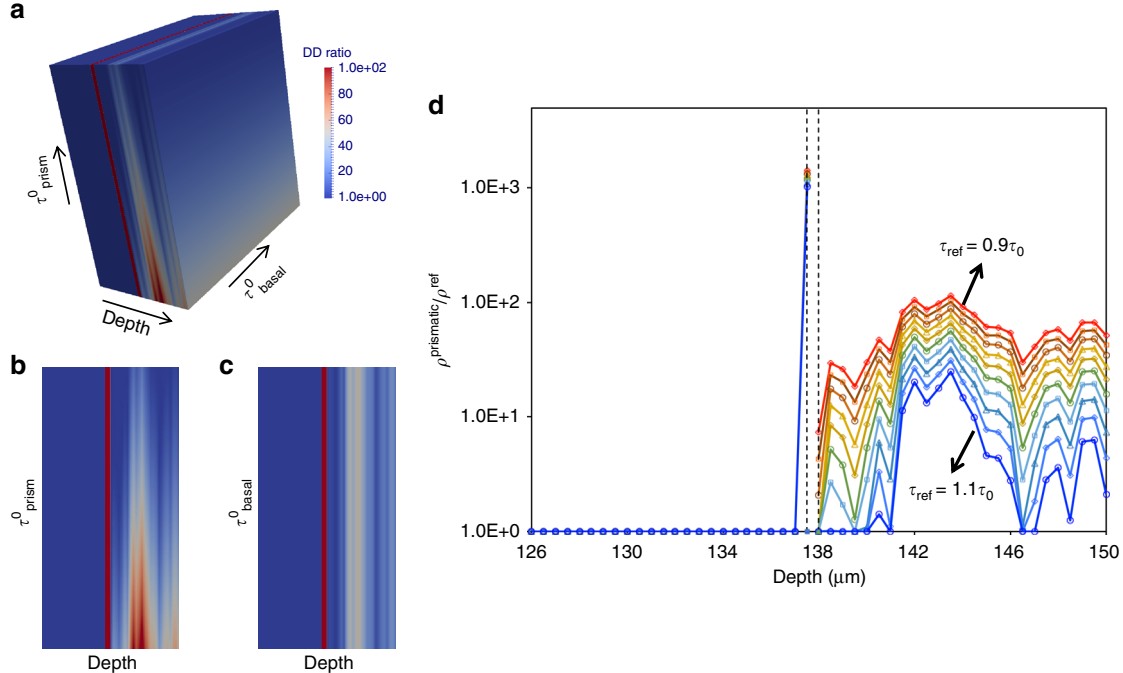

**Fig 5** Asymmetric distribution of prismatic dislocation density ratio. **a** Calculated prismatic slip-1 dislocation density ratio (=$\rho^{Prismatic}/\rho^{ref}$) profile for different basal and prismatic modes threshold shear stress values. The range for threshold stress values of basal and prismatic modes is ±10% of its CRSS. **b**, **c** DD ratio profile for fixed basal and prismatic modes threshold shear stress values (=actual CRSS), respectively. **d** The distribution of DD ratio for different prismatic threshold shear stress values while fixing basal threshold shear stress value. The reference dislocation density is $1 \times 10^{12}\,\mathrm{m}^{-2}$

show the distribution of dislocation density ratio along line A-B for a set of selected threshold stress values of prismatic slip in Fig. 5d. The dislocation density ratio is high for low reference value and vice versa.

**Implications of the study and directions for future research.** Our aim in this study is to measure the local stresses in the vicinity of a twin within a bulk sample of Mg alloy AZ31. Using X-ray synchrotron with differential-aperture X-ray microscopy technique, we have successfully demonstrated the procedure to measure the spatially resolved, complete stress and strain fields across a twin boundary. To understand and interpret the measurements, we have employed a generalized Taylor law to quantify the local dislocation density distribution for different slip systems. Although the measurement refers to the final state of the twinned grain, this experiment allows us to reconstruct the sequence of micromechanical events that takes place in a grain when a twin partitions it. The sequence is: the selected grain deforms elastically first, twinning splits the grain into two domains. In subsequent straining steps, one of the domains deforms elastically and the other undergoes prismatic slip plasticity. A preliminary conclusion is that grain partitioning by twinning creates two sub-grains that evolve stress, strain, and microstructure differently, and suggests that such situation should be included in modeling.

The observed splitting of a grain into non-interacting domains by twinning is interesting in itself, but for it to promote a better understanding of twinning, the following questions should be answered: Are the observed micromechanical processes statistically relevant; and what mechanism governs the observed variation in the stress field at each side of the twin? We speculate that the interaction with neighbors creates stress variations in the grain, which are responsible for driving the relaxation mechanism represented by the twin, and for the subsequent partition into elastic and plastic domains. In addition, it has to be kept in mind

that twin expansion is a three-dimensional process, and that the configuration of neighboring grains is also three-dimensional. Ideally, with unlimited access to X-ray diffraction, one could start addressing both questions by repeating this experiment for several twins in three-dimension while simultaneously characterizing the neighborhood. This would significantly advance the understanding of twinning in 3D and also guide the modeling tool development. In this work, the major part of the beam time was spent identifying a grain with a twin and to obtain basic details like grain lattice orientation, grain position and twin type. Only after this we were able to measure local strains. Thus, a potential future experiment characterizing several twins in 3D is possible if we were able to effectively identify and obtain such previous basic information for a set of twins. Recently Abdolvand et al.[42] and Bieler et al.[7] successfully demonstrated that 3D-XRD is a powerful tool for determining the center-of-mass, lattice orientation, average elastic strain and average stress for a large number of grains and twins in a polycrystal. Using the center-of-mass they identified the neighboring orientations for all individual grains and twins, and studied the role of neighboring grains on individual grain deformation response. Note that these works can only provide the grain average responses, not the spatially resolved strain and stress distributions presented in this work. We suggest combining the present experimental method (DAXM technique) with 3D-XRD as an effective way to measure the local stresses for several twins in 3D and to establish the effect of neighboring grains on twin local stresses.

## Methods

**Differential aperture scanning microscopy.** The rectangular specimen was cut from the rolled plate and compressed along the RD favoring activation of easy {10$\bar{1}$2} tensile twinning. The incident X-ray beam was aligned at 45° to the TD axis. The micro-diffraction measurements were made after applying an initial compression under fixed applied load. The spatially resolved 3D micro-diffraction measurements were conducted by using differential-aperture X-ray microscopy (DAXM) at beamline 34-ID-E of the Advanced Photon Source in Argonne National Laboratory. An X-ray microbeam (polychromatic or monochromatic) was

focused to about 0.5 μm × 0.5 μm by a pair of elliptical Kirkpatrick–Baez focusing mirrors. Diffracted X-ray beams are recorded on X-ray area detectors. The depth resolution is provided by scanning a 100 μm-diameter platinum wire as diffracted-beam profiler in a knife-edge fashion. By triangulation, the origins of the X-rays can be determined on a pixel-by-pixel basis. A typical scanning step of 0.7 μm gives the depth resolution of ~0.5 μm. By scanning sample along rolling direction (RD), an incident X-ray slicing plane at 45° to the sample surface can be measured; hence a map of the crystal orientation in polychromatic beam mode is obtained. In the monochromatic beam mode, the energy is scanned through the Bragg peak in 2 eV steps. By combining data from all of the energies and wire positions, the X-ray intensity at each pixel on the detector was determined as a function of photon energy at each depth step. Since we know the locations of the pixels on the detector relative to the incident beam, the diffraction angle can be calculated and converted to diffraction vector $\mathbf{q}$ in reciprocal $k$-space. Three non-coplanar diffraction vectors are required to determine the full strain tensor.

**3D fitting of diffraction peaks**. The measured set of diffracted beam images at different energies comprises a sampling of $k$-space around each measured Bragg peak. This non-uniform sampling of $k$-space can be easily binned into a uniform sampling in $\{q_x, q_y, q_z\}$ centered on the prominent Bragg peak. This peak was then fitted to a 3D Gaussian distribution, the exponential function

$$A \, \exp\left[-\left(\frac{q_x - q_{cx}}{2\sigma_x}\right)^2 - \left(\frac{q_y - q_{xy}}{2\sigma_y}\right)^2 - \left(\frac{q_z - q_{cz}}{2\sigma_z}\right)^2\right] \qquad (2)$$

to determine $\mathbf{q}_c = \{q_{cx}, q_{cy}, q_{cz}\}$, the center of observed Bragg peak in all 3 dimensions. With 3 non-coplanar $\mathbf{q}$'s and the associated hkl's it is a simple matter of applying linear algebra to obtain all 9 components of the strained lattice vectors.

## Data availability

The data that support the findings of this study are available from the corresponding author upon reasonable request.

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

## Acknowledgements

This work was fully funded by the U.S. Dept. of Energy, Office of Basic Energy Sciences Project FWP 06SCPE401. Diffraction experiments were performed at the Advanced Photon Source, supported by the US Department of Energy, Office of Basic Energy Sciences, under Contract No. DE-AC02-06CH11357.

## Author contributions

Experiments are performed by M.A.K, B.C, W.L, C.N.T, L.C and R.J.M. Post-processing of diffraction data are performed by J.T, B.C and M.A.K. Results are discussed and interpreted by M.A.K, L.C, C.N.T and B.C. All authors commented on the manuscript.

## Additional information

**Competing interests:** The authors declare no competing interests.

