## [Peer Review File · Nature Communications]

Reviewers' comments:

Reviewer #1 (Remarks to the Author):

In this paper, the state of deformation at the vicinity of a twin in a magnesium grain is studied in detail. Twinning is an important mode of deformation in Hexagonal Close-Packed (HCP) polycrystals such as Magnesium, Zirconium or Titanium. Therefore, although experiment was conducted on a magnesium sample, the paper will have broad audience since it is a fundamental study.

I am familiar with the work conducted by this group, particularly by Drs. Carlos Tome, Clausen and Capulungo, and more recently Kumar. The team has been tackling some of the most interesting, and perhaps difficult aspects, of deformation twinning. They claim that they have conducted the very first in-situ synchrotron study with differential aperture X-ray microscopy (DAXM). In contrast to 3D-XRD where a large volume of the polycrystal can be probed to find twins, with DAXM, just a few grains can be probed with the advantage of having many measurement points per grain. This means that during the in-situ experiment, one should primarily focus on finding a grain that twins- For a given beamtime, this can be very stressful and sometime impossible to find the perfect grain. Given the difficulty of doing this experiment, and the fact that this is the first report, I think the paper merits publication in this journal, after addressing the following comments:

(1) The biggest missing part of this paper is that I couldn't find the in-situ stress-strain curve for the sample. Authors need to add this to their paper. Also, it is not clear at what macroscopic strain/stress the measurement on the green grain (the parent grain) was conducted, when it actually twinned and to what extent they followed deformation within this grain. This is central to the study.

(2) Authors have claimed that once twinning happened the parent grain splits in two with one side deforming elastically and one side plastically, but there is no data to support this, so as for before twinning; they have measured it but probably due to lack of space did not add the data. If there is a problem with space I recommend them to prepare a strong supplementary document and add these figures (and the following ones- see below). On this stress-strain curve (from comment 1) please mark where the parent grain twins, at what strain you started looking at it, and to what strain you actually measured localized stress fields around the twin. This will allow other members of the community to understand this work better and perhaps, replicate it for their model development.

(3) They need to clarify how they calibrated the single crystal parameters of the Mg sample. Is this measured at no applied load? Since lattice strains are quite small for Mg, any small variation in D0 will give very different results for the both sides of the twin and will change the conclusions made. Some papers are referenced (by Levente Balogh) but it is necessary to comment on this. Do they use the D0 measured at SMARTS as well?

(4) In the abstract and in the paper, authors claim that this is the first in-situ experiment with DAXM on twins; I couldn't find how different their experiment is comparing to Reference 48, By Balogh et al, Acta Mater, 61, 2013, 3612. In this reference, DAXM was used at the vicinity of twins in MgAz31 and they showed stress gradient close to a twin boundary as well as within a twin. Current authors need to highlight what they did differently comparing to this reference and what is new in their work.

(5) Regarding the stress fields close to twin boundary: The main question to ask is that if the line A-B is moved a bit toward north or south (on TD-RD plane in the RD direction), do authors see the same trends? There are a couple of HR-EBSD studies on other HCP polycrystals showing similar

trends to this paper where it is shown that the stress along the line A-B changes very much when it is moved up and down. This is also observed in crystal plasticity modelling- and I agree that it is very much a reflection of the neighbouring grains as well as the state of twinning (nucleation and propagation as oppose to thickening).

(6) Authors have commented on Hall-Petch effects and the fact that after twinning the parent grain will split in two distinct sub grains that deform independently. If they were to start with a much bigger grain with much bigger sub grains after twinning, would the conclusions stand? There are so many parameters that can affect stress along A-B and I think this group is in solid position to comment on that using FFT modelling. If they import the measured micro-structure into their FFT code and model the twin, do they not see the same stress field that they captured along A-B? I understand if they don't go with modelling at this stage, but they will need to comment on if the current CP models can actually capture such stress variations along a typical A-B line (parent-twin pairs) and if not, why not and what should be done?

(7) Measurements were made every 0.5 microns. What was the X-ray beam size? Was it a square beam?

(8) It is not easy to separate each components of stress tensor shown in fig2, but there seem to be a huge hydrostatic stress at the left hand side of the parent grain- is there any reason for this?

(9) Due to experimental set-up it was not possible to measure stress within twin, however, have the authors looked into the traction force acting normal to the twin habit plane at the two sides of the parent grain? Is the traction force continuous?

(10) Line 228-231: What was the value of the lattice rotation that they measured? Can they plot the W12, 13, and 23 contours? This will be very valuable.

(11) Fig. 4b- τ_{av_0} is plotted with $\pm 10\%$ variation- Question: how does the prismatic slip system harden in MgAz31? Do they not harden or slightly harden?

(12) Dislocation densities and prismatic sensitive plasticity: I am not sure if I can follow what authors want to say; extracting dislocation density from a hardening law. If they want to discuss dislocation densities, can they not just take a look at the peak broadening from their experiment? Another question: How far were the three diffractometer from the sample? These are the missing information that I needed to understand why they picked up Eq. 1 as oppose to other possible methods...

(13) Lines 327-329: Have they looked at the sample surface to confirm the activity of prism slip system (by formation of slip bands) or is this based on diffraction patterns?

Minors:

It is not possible to distinguish S11 from S12 or the other stress components shown in Fig. 2. Please re-plot using dashed line, etc

Reviewer #2 (Remarks to the Author):

The paper reported a technique which measures the 3D stress field with high spatial resolution and highlighted the advantage over other 3D-XRD methods which only give the average stress and strain in a grain. The method is definitely promising and attractive to the plasticity community and

materials community at large where local stress state is of interest. The reviewer has the following comments and questions for clarification and further explanation. Specifically, the reviewer is confused by the discussion of the sequence of plastic events involved with twinning.

(1) To the reviewer's understanding, in the work of ref. 48, the method has been comprehensively applied to AZ31 alloy with both poly- and mono-chromatic X-rays. The cited work has also achieved spatially resolved measurement of strain in a crystallographic direction and has shown different strains in domains separated by twins. The reviewer would like to ask the current authors to highlight the advancement of the current work.

(2) Please clarify, by stating in-situ experiment, is the sample firstly compressed and then hold during the measurement (line 116)?

(3) In line 130, it states "a stable [10-12] tensile twin". Does it mean specifically a (10-12) twin or generally a {10-12} type twin? The question may seem demanding, but subsequently in line 289, a specific prismatic system is identified. One wonders, what is the coordinate system of the matrix and twin and what is special about this one prismatic slip system with respect to the twinning plane?

(4) In Fig. 2 and 3, it is emphasized that the significantly different stress value occurs only on the left side of the TB but not on the right. Since the zone is approaching the resolution of the measurement, one would argue whether the asymmetry is an intrinsic material response or is due to the resolution. Does it also occur on the right side of the TB but is not captured? Before confirming this result, it is tenuous to comment on the asymmetric twin growth. In fact, in Fig. 3, the first couple of data points on either side of the twin have very similar values. It seems to suggest that the near field stress at the two TBs are similar.

(5) Analyses of the RSS. While the reviewer does not question the computed RSS values from the measured full stress tensor and agrees with the variation in space and that they are influenced by the surrounding grain neighborhood (Fig. 4), what the reviewer cannot reconcile is the fact that the CRSS of prismatic slip has been reached. If so, why is the system not activated at the moment? In other words, how can the elastic stress exceed the yield stress? The measured and hence computed stresses are at load holding, i.e. after plasticity, wouldn't high elastic stress suggest either hardening or the inability of the region to plastically relax the imposed stress/strain?

(6) It is further inferred that with additional increment of strain, the parent to the right of the twin likely deforms by prismatic slip and the parent to the left of the twin only deforms elastically (line 226 to 228). If the parent grain is subject to compressive stress in direction 2, as stated in line 183 and 184, by simple Schmid law, one would expect the grain is suitably oriented for prismatic slip. When the global stress increases during subsequent loading, why would one domain deform plastically but the other not? Without the information on the evolution of the stress – strain during loading, the reviewer does not see a clear link between the current stress and the subsequent stress or even the previous stress, and therefore does not see the basis for the inference. A similar statement appears in line 248 to 250.

(7) The statements of lattice rotation in line 229 and the prediction of severe deformation and hardening in line 231 and 232 are not supported by evidence. Are these findings or hypotheses?

(8) Line 257, the sentence seems incomplete.

(9) Line 258 to 259. Doesn't the internal stress/strain evolution also inform the sequence of deformation events? Maybe the emphasis of the current technique is spatially resolved? It would be nice if the authors could elucidate the distinctions to enlighten the readers.

(10) The issues in (5) and (6) also affect the section from line 268 to 307.

By using the Taylor law, the higher threshold value is attributed to strain hardening. Going back to the points of (5) and (6), if prismatic is hardened so much on the right of the twin, during subsequent straining, why is prismatic slip expected on the right but not on the left?

Response to Reviewer - 1

In this paper, the state of deformation at the vicinity of a twin in a magnesium grain is studied in detail. Twinning is an important mode of deformation in Hexagonal Close-Packed (HCP) polycrystals such as Magnesium, Zirconium or Titanium. Therefore, although experiment was conducted on a magnesium sample, the paper will have broad audience since it is a fundamental study.

I am familiar with the work conducted by this group, particularly by Drs. Carlos Tome, Clausen and Capulungo, and more recently Kumar. The team has been tackling some of the most interesting, and perhaps difficult aspects, of deformation twinning. They claim that they have conducted the very first in-situ synchrotron study with differential aperture X-ray microscopy (DAXM). In contrast to 3D-XRD where a large volume of the polycrystal can be probed to find twins, with DAXM, just a few grains can be probed with the advantage of having many measurement points per grain. This means that during the in-situ experiment, one should primarily focus on finding a grain that twins- For a given beamtime, this can be very stressful and sometime impossible to find the perfect grain. Given the difficulty of doing this experiment, and the fact that this is the first report, I think the paper merits publication in this journal, after addressing the following comments:

Response: We thank the referee for the detailed review of our manuscript. We found the comments valuable toward improving it. Following his/her suggestions we added in the appropriate places the requested clarifications and discussions.

(1) The biggest missing part of this paper is that I couldn't find the in-situ stress-strain curve for the sample. Authors need to add this to their paper. Also, it is not clear at what macroscopic strain/stress the measurement on the green grain (the parent grain) was conducted, when it actually twinned and to what extent they followed deformation within this grain. This is central to the study.

Response: In this work, the sample is pre-compressed on a vise, and the measurement is performed under strain hold. We do not perform the actual in-situ experiment and thus we cannot provide the in-situ stress-strain curve. At the same time, we agree that the term 'in-situ' leads to confusion, and we have replaced 'in-situ' with 'under applied load' in the revision. In addition, the measurement is performed in the bulk of a polycrystalline sample and the individual grain stress state is unlikely to be the same as the macroscopic one. As a consequence, the macroscopic stress-strain response will not provide insight into the measured local stress distributions.

(2) Authors have claimed that once twinning happed the parent grain splits in two with one side deforming elastically and one side plastically, but there is no data to support this, so as for before twinning; they have measured it but probably due to lack of space did not add the data. If there is a problem with space I recommend them to prepare a strong supplementary

document and add these figures (and the following ones- see below). On this stress-strain curve (from comment 1) please mark where the parent grain twins, at what strain you started looking at it, and to what strain you actually measured localized stress fields around the twin. This will allow other members of the community to understand this work better and perhaps, replicate it for their model development.

Response: As mentioned above, we have not measured in-situ the stress evolution leading to twin activation in the selected grain. However, although the measurement is performed under constant applied load, we propose a plausible sequence of events, as follows. The selected grain contains only one twin, and it has been shown by in-situ work of Aydiner et al. (2009) in AZ31 Mg alloy that the parent deforms elastically, until twinning is activated in a grain at a stress of ~ 75 MPa, before the parent shows sign of plastic yielding (see Fig below). And so we conclude that in Mg alloy the twin forms in an elastically deforming grain.

Figure: Evolution of S33 component as a function of applied stress in the parent grain and in the twin domains in HCP AZ31 Mg alloy.

(3) They need to clarify how they calibrated the single crystal parameters of the Mg sample. Is this measured at no applied load? Since lattice strains are quite small for Mg, any small variation in d_0 will give very different results for the both sides of the twin and will change the conclusions made. Some papers are referenced (by Levente Balog) but it is necessary to comment on this. Do they use the d_0 measured at SMARTS as well?

Response: Thanks for pointing this out. We have calibrated the single crystal parameters of the Mg sample at stress free condition using the same 34-ID-E beamline of the Advanced Photon Source. We have not measured d_0 at SMARTS. In the revised manuscript, we describe how we get the initial single crystal parameters.

(4) In the abstract and in the paper, authors claim that this is the first in-situ experiment with DAXM on twins; I couldn't find how different their experiment is comparing to Reference 48, By Balogh et al, Acta Mater, 61, 2013, 3612. In this reference, DAXM was used at the vicinity of twins in MgAz31 and they showed stress gradient close to a twin boundary as well as within a twin. Current authors need to highlight what they did differently comparing to this reference and what is new in their work.

Response: We agree that the current version of the paper doesn't compare the present work with Ref. 48. Both the present work and Ref. 48 use the same experimental technique at APS 34-ID-E (X-ray synchrotron facility with differential-aperture X-ray microscopy) to measure the elastic strain in the vicinity of a tensile twin.

In Ref. 48, we used only one diffraction vector, which gives us only one elastic strain component, namely, the strain perpendicular to a crystallographic plane in the parent and in the twin. Using this arbitrary elastic strain variation along a line that crosses the grain and the twin, we could only conclude that the elastic strain field is non-homogeneous in the vicinity of the twin. The experiment in the current work, on the other hand, consists on determining the elastically deformed unit cell of the crystal at each local point using 3 different monochromatic X-ray frequencies. The distortion of the unit cell allows us to calculate the six elastic components of the local strain tensor, and so to map the full local stress tensor.

In addition to the advancement in the experimental technique and post-process, the present work is significantly different from Ref. 48 in that it allows for microstructural analysis: we are able to calculate local dislocation density in the vicinity of the twin and local crystal rotations. In the revised version, we address this point in the introduction by adding the following text:

Recently, experimental and modeling efforts were devoted to characterize the twin local stresses and the role of neighbors (Abdolvand and Wilkinson, 2016a; Abdolvand and Wilkinson, 2016b; Abdolvand et al., 2018; Ardeljan et al., 2015; Balogh et al., 2013; Basu et al., 2018; Bieler et al., 2014; Kumar et al., 2015; Wang et al., 2010; Wang et al., 2013). Among all the experimental literature, only the work of Balogh et al. focuses on the local elastic strain within and in the vicinity of a twin inside a bulk polycrystalline material. But they only measured one component of elastic strain, along an arbitrarily chosen diffraction vector. While this procedure shows the presence of a gradient in the vicinity of

the twin, it does not allow one to derive the full elastic strain tensor and the associated stress tensor. Knowledge of the latter is essential to extract meaningful conclusions from the experiment and to assess the effect of local stress distribution on further twin thickening.

(5) Regarding the stress fields close to twin boundary: The main question to ask is that if the line A-B is moved a bit toward north or south (on TD-RD plane in the RD direction), do authors see the same trends? There are a couple of HR-EBSD studies on other HCP polycrystals showing similar trends to this paper where it is shown that the stress along the line A-B changes very much when it is moved up and down. This is also observed in crystal plasticity modelling- and I agree that it is very much a reflection of the neighbouring grains as well as the state of twinning (nucleation and propagation as oppose to thickening).

Response: If we move the line A-B a bit toward north or south within the middle of the grain, we do not expect any changes in the findings. The absolute stress values may slightly change, but the overall response will be the same. At the same time, if we move closer to the grain boundaries, then we expect to get different results, which will be mediated by the neighboring grains. In this work we are interested in the local stresses associated with twinning. That is why we have performed the measurement in the middle of the grain, not close to the grain boundaries.

(6) Authors have commented on Hall-Petch effects and the fact that after twinning the parent grain will split in two distinct sub grains that deform independently. If they were to start with a much bigger grain with much bigger sub grains after twinning, would the conclusions stand? There are so many parameters that can affect stress along A-B and I think this group is in solid position to comment on that using FFT modelling. If they import the measured micro-structure into their FFT code and model the twin, do they not see the same stress field that they captured along A-B? I understand if they don't go with modelling at this stage, but they will need to comment on if the current CP models can actually capture such stress variations along a typical A-B line (parent-twin pairs) and if not, why not and what should be done?

Response: In this work we have observed the twin induced splitting of the grain into non-interacting domains. At the same time, we are not saying that the twins will always split the grain into non-interacting domains. In the manuscript we also ponder whether the observed grain partitioning is statistically relevant or not, an issue which we plan to address in the future.

We agree that the measured stress distribution and the observed grain partitioning will depend on several factors. To interpret and explain the observed stress variations, first we need complete 3D microstructural information, but we don't have it now. Without that we cannot perform the FFT simulation to check whether it captures the observed stress fields. This point is mentioned in the manuscript as future research, and we are in

the process of measuring local stresses in the vicinity of a 3D twin in a 3D microstructure, as a follow up on the present work where measurements are performed in a particular section.

(7) Measurements were made every 0.5 microns. What was the X-ray beam size? Was it a square beam?

Response: Yes, it is a square beam of 0.5 micron size and it is reported in the revision.

(8) It is not easy to separate each components of stress tensor shown in fig2, but there seem to be a huge hydrostatic stress at the left hand side of the parent grain- is there any reason for this?

Response: To explain the hydrostatic pressure measured in the vicinity of the twin we need a complete 3D microstructure of twin and grain. However we can confirm that the values are reliable because this point has sufficient diffraction counts for line profile analysis. In addition, our own simulations of shear transformation for an oblate tensile twin in Mg, show large localizations of the three diagonal stress components in the vicinity of the twin interface. At several points in the vicinity of twin, the three diagonal stress components have the same sign and lead to large hydrostatic stress. This provides further feasibility to the measurement (see Figure below).

Figure: s_{11} , s_{22} and s_{33} stress components induced by shear transformation of an oblate tensile twin in Mg (continuum simulations courtesy H. Tummala). Units of MPa.

(9) Due to experimental set-up it was not possible to measure stress within twin, however, have the authors looked into the traction force acting normal to the twin habit plane at the two sides of the parent grain? Is the traction force continuous?

Response: The studied twin thickness is ~ 0.5 micron, which is the resolution of our experiment. It leads to the omission of one data point that cover most part of the twin and a small part of the parent grain on the right side of the twin. And so we cannot study stresses inside the twin or the traction force normal to twin interface.

(10) Line 228-231: What was the value of the lattice rotation that they measured? Can they plot the W12, 13, and 23 contours? This will be very valuable.

Response: We have calculated the lattice rotation using the average crystal orientation of the left and right side of the grain, and those values are: $W_{12} = 0.0345$; $W_{13} = 0.0005$; and $W_{23} = 0.0022$, which corresponds to a 2.0 degree rotation of the crystal about $[0.0633; -0.0017; 0.9980]$ direction. This particular lattice rotation can be achieved only by the activation of (01-10)[2-1-10] prismatic slip. The lattice rotations are reported in the revised manuscript.

(11) Fig. 4b- tau_0 is plotted with +/-10% variation- Question: how does the prismatic slip system harden in MgAz31? Do they not harden or slightly harden?

Response: The +/-10% variation represents a local fluctuation of the initial critical resolved shear stress due to dispersion in the intra-granular microstructure. The band is meant to show that within a 10% fluctuation, basal and pyramidal slip will not be activated.

(12) Dislocation densities and prismatic sensitive plasticity: I am not sure if I can follow what authors want to say; extracting dislocation density from a hardening law. If they want to discuss dislocation densities, can they not just take a look at the peak broadening from their experiment? Another question: How far were the three diffractometer from the sample? These are the missing information that I needed to understand why they picked up Eq. 1 as oppose to other possible methods...

Response: The equation (1) is a way to extract the individual dislocation densities by assuming that the local resolved shear stress (which is inferred from the measurement) evolves during slip in synch with the local Critical Resolved Shear Stress. We agree that the dislocation densities can also be calculated by peak broadening analysis, but we do not have enough counts to resolve the peak profile and perform peak broadening analysis.

(13) Lines 327-329: Have they looked at the sample surface to confirm the activity of prism slip system (by formation of slip bands) or is this based on diffraction patterns?

Response: Our measurement is performed inside the bulk polycrystalline materials, and so we cannot use the information from the sample surface to confirm the activation of prismatic slip system.

Minors:

It is not possible to distinguish S11 from S12 or the other stress components shown in Fig. 2. Please re-plot using dashed line, etc

Response: Thanks for pointing out. We have re-plotted using dashed line and different marker styles.

Response to Reviewer - 2

The paper reported a technique which measures the 3D stress field with high spatial resolution and highlighted the advantage over other 3D-XRD methods which only give the average stress and strain in a grain. The method is definitely promising and attractive to the plasticity community and materials community at large where local stress state is of interest. The reviewer has the following comments and questions for clarification and further explanation. Specifically, the reviewer is confused by the discussion of the sequence of plastic events involved with twinning.

Response: We thank the referee for the careful and detailed review of our manuscript.

(1) To the reviewer's understanding, in the work of ref. 48, the method has been comprehensively applied to AZ31 alloy with both poly- and mono-chromatic X-rays. The cited work has also achieved spatially resolved measurement of strain in a crystallographic direction and has shown different strains in domains separated by twins. The reviewer would like to ask the current authors to highlight the advancement of the current work.

Response: We agree that the current version of the paper doesn't compare the present work with Ref. 48. Both the present work and Ref. 48 use the same experimental technique at APS 34-ID-E (i.e., X-ray synchrotron facility with differential-aperture X-ray microscopy) to measure the elastic strain in the vicinity of a tensile twin.

In Ref. 48, we used only one diffraction vector, which gives us only one elastic strain component, namely, the strain perpendicular to a crystallographic plane in the parent and in the twin. Using this arbitrary elastic strain variation along a line that crosses the grain and the twin, we could only conclude that the elastic strain field is non-homogeneous in the vicinity of the twin. The experiment in the current work, on the other hand, consists on determining the elastically deformed unit cell of the crystal at each local point using 3 different monochromatic X-ray frequencies. The distortion of the unit cell allows us to calculate the six elastic components of the local strain tensor, and so to map the full local stress tensor.

In addition to the advancement in the experimental technique and post-process, the present work is significantly different from Ref. 48 in that it allows for microstructural analysis. We are able to calculate local dislocation density in the vicinity of the twin using Taylor's law. In the revised version, we address this point in the introduction by adding the following text:

Recently, experimental and modeling efforts were devoted to characterize the twin local stresses and the role of neighbors (Abdolvand and Wilkinson, 2016a; Abdolvand and Wilkinson, 2016b; Abdolvand et al., 2018; Ardeljan et al., 2015; Balogh et al., 2013; Basu et al., 2018; Bieler et al., 2014; Kumar et al., 2015; Wang et al., 2010; Wang et al., 2013). Among all the experimental literature, only the work of Balogh et al. focuses on the local elastic strain within and in the vicinity of a twin inside a bulk polycrystalline material. But they only measured one component of elastic strain, along an arbitrarily chosen diffraction vector. While this procedure shows the presence of a gradient in the vicinity of the twin, it does not allow one to derive the full elastic strain tensor and the associated stress tensor. Knowledge of the latter is essential to extract meaningful conclusions from the experiment and to assess the effect of local stress distribution on further twin thickening.

(2) Please clarify, by stating in-situ experiment, is the sample firstly compressed and then hold during the measurement (line 116)?

Response: Thanks for pointing this out. Yes, we first compress the sample and then the measurements are done under hold. We agree that our experiment is not 'in-situ'. To avoid the confusion we have replaced the term 'in-situ' with 'under applied load' in the revised manuscript.

(3) In line 130, it states "a stable [10-12] tensile twin". Does it mean specifically a (10-12) twin or generally a {10-12} type twin? The question may seem demanding, but subsequently in line 289, a specific prismatic system is identified. One wonders, what is the coordinate system of the matrix and twin and what is special about this one prismatic slip system with respect to the twinning plane?

Response: Thanks for pointing out the inconsistency. The exact crystallography of the activated twin is (0-112)[01-11] and it is updated in the revised manuscript. The active prismatic slip system is not favorably oriented to activate by the same shear stress that activates the twin. In this case, we speculate that the back stress induced by the twinning transformation, plus the reaction exerted by the surrounding grains mediate the activation of the prismatic slip system.

(4) In Fig. 2 and 3, it is emphasized that the significantly different stress value occurs only on the left side of the TB but not on the right. Since the zone is approaching the resolution of the measurement, one would argue whether the asymmetry is an intrinsic material response or is due to the resolution. Does it also occur on the right side of the TB but is not captured? Before confirming this result, it is tenuous to comment on the asymmetric twin growth. In

fact, in Fig. 3, the first couple of data points on either side of the twin have very similar values. It seems to suggest that the near field stress at the two TBs are similar.

Response: We agree that the size of the twin-affected zone is within the resolution of our measurements. At the same time, we are not claiming the possibility for asymmetric twin growth by comparing the stress values of a point on each side of the twin. Instead, we use the observed asymmetry in the slip activity (Fig. 4) and in the dislocation density (Fig. 5) to propose the possibility for asymmetric twin growth. To clarify this point, we have added the following text in the revised manuscript in page. 9:

The heterogeneous deformation behavior of the twinned grain likely has important implications on mechanical behavior and further twinning. Particularly, the asymmetry in the plastic slip activity and in the dislocation population between left and right part of the grain may lead to asymmetric twin thickening under further straining.

(5) Analyses of the RSS. While the reviewer does not question the computed RSS values from the measured full stress tensor and agrees with the variation in space and that they are influenced by the surrounding grain neighborhood (Fig. 4), what the reviewer cannot reconcile is the fact that the CRSS of prismatic slip has been reached. If so, why is the system not activated at the moment? In other words, how can the elastic stress exceed the yield stress? The measured and hence computed stresses are at load holding, i.e. after plasticity, wouldn't high elastic stress suggest either hardening or the inability of the region to plastically relax the imposed stress/strain?

Response: First of all, we agree that the elastic stress can't exceed the yield stress, and so the RSS is always smaller or equal than the CRSS. In the analysis we mean to show that the initial CRSS, which corresponds to the stress-free condition, has been exceeded. Note that the CRSS value will evolve with strain hardening. The RSS values are calculated for the final state in the strained material, and so the prismatic slip system may be activated and hardened before our measurement. That is the reason the RSS value of a prismatic slip system is greater than the initial CRSS value. This analysis helps to check whether any particular slip system has been activated or not. That is, if the RSS value of the strained material is greater than the initial CRSS, then we can confirm that the particular slip system was activated.

In the manuscript, we do not state explicitly that the used CRSS values correspond to initial stress free condition, and this is likely the reason for the reviewer's concern. It is addressed in the revised manuscript in page 9 as,

In these figures, the reference line represents the critical resolved shear stress (CRSS) required to activate the slip. These reference stress values correspond to stress-free initial conditions.

(6) It is further inferred that with additional increment of strain, the parent to the right of the twin likely deforms by prismatic slip and the parent to the left of the twin only deforms elastically (line 226 to 228). If the parent grain is subject to compressive stress in direction 2,

as stated in line 183 and 184, by simple Schmid law, one would expect the grain is suitably oriented for prismatic slip. When the global stress increases during subsequent loading, why would one domain deform plastically but the other not? Without the information on the evolution of the stress – strain during loading, the reviewer does not see a clear link between the current stress and the subsequent stress or even the previous stress, and therefore does not see the basis for the inference. A similar statement appears in line 248 to 250.

Response: We agree with the reviewer that using the available stress distribution is not possible to comment on the activation of slip systems for further straining. Under further straining, based on the constraints experienced, each part of the grain may or may not activate slip systems. At the same time, we can claim that the selected grain has undergone asymmetric slip activity. In accordance with that we have updated the line 226 to 228 as,

In summary, the parent to the right of the twin deforms plastically by prismatic slip, whereas the parent to the left of the twin only deforms elastically.

(7) The statements of lattice rotation in line 229 and the prediction of severe deformation and hardening in line 231 and 232 are not supported by evidence. Are these findings or hypotheses?

Response: Both, the lattice rotation and local deformation, are extracted from the measurement, not hypothesis. At the same time, we agree that we are not providing the calculated lattice rotation. We do now in the revision as,

We estimate the latter by comparing the average orientation of the voxels on the left and right side of the twin, and the calculated average lattice rotation is: $W_{12} = 0.0345$; $W_{13} = 0.0005$; and $W_{23} = 0.0022$, which corresponds to a 2.0 degrees rotation of the crystal about $[0.0633; -0.0017; 0.9980]$ direction. This particular lattice rotation can be achieved by the activation of $(01-10)[2-1-10]$ prismatic slip.

(8) Line 257, the sentence seems incomplete.

Response: Thanks for pointing out. Now it reads,

After twinning, the grain domain on the left side deforms elastically and the one on the right side deforms plastically.

(9) Line 258 to 259. Doesn't the internal stress/strain evolution also inform the sequence of deformation events? Maybe the emphasis of the current technique is spatially resolved? It would be nice if the authors could elucidate the distinctions to enlighten the readers.

Response: We agree with the reviewer that (potentially) this technique could be used to follow in-situ internal stress/strain evolution, and to identify the sequence of events. The beam-time required, however, would be prohibitively long (our experiment required 18 shifts of 8 hours each). As the reviewer points out, the value of our experiment resides in

the spatial resolution of stress/strain fields in a bulk grain, for a fixed microstructure. Our only safe speculation is that the grain characterized must have been twin-free and deforming elastically up to when the twin transformation took place.

(10) The issues in (5) and (6) also affect the section from line 268 to 307. By using the Taylor law, the higher threshold value is attributed to strain hardening. Going back to the points of (5) and (6), if prismatic is hardened so much on the right of the twin, during subsequent straining, why is prismatic slip expected on the right but not on the left?

Response: The upper bound threshold value used in the Taylor law is not attributed to strain hardening. The $\pm 10\%$ variation represents the fluctuation in the threshold value due to intra-granular microstructure. It is already mentioned in page 11.

At the same time, we agree with the reviewer that using the available stress distribution is not possible to comment on the slip activity during subsequent straining, and it addressed in the revised manuscript. However, at the state of measurement, we can confirm that the left of the twin is elastic, because the RSS values of prismatic slip systems in the strained material are smaller than the initial CRSS.

References:

- Abdolvand, H., Wilkinson, A., 2016a. On the effects of reorientation and shear transfer during twin formation: Comparison between high resolution electron backscatter diffraction experiments and a crystal plasticity finite element model. *Int J Plasticity* 84, 160-182.
- Abdolvand, H., Wilkinson, A.J., 2016b. Assessment of residual stress fields at deformation twin tips and the surrounding environments. *Acta Mater* 105, 219-231.
- Abdolvand, H., Wright, J., Wilkinson, A., 2018. Strong grain neighbour effects in polycrystals. *Nat Commun* 9.
- Ardeljan, M., McCabe, R.J., Beyerlein, I.J., Knezevic, M., 2015. Explicit incorporation of deformation twins into crystal plasticity finite element models. *Comput Method Appl M* 295, 396-413.
- Balogh, L., Niezgodá, S.R., Kanjarla, A.K., Brown, D.W., Clausen, B., Liu, W., Tome, C.N., 2013. Spatially resolved in situ strain measurements from an interior twinned grain in bulk polycrystalline AZ31 alloy. *Acta Mater* 61, 3612-3620.
- Basu, I., Fidler, H., Ocelik, V., de Hosson, J.T.M., 2018. Local Stress States and Microstructural Damage Response Associated with Deformation Twins in Hexagonal Close Packed Metals. *Crystals* 8.
- Bieler, T., Wang, L., Beaudoin, A., Kenesei, P., Lienert, U., 2014. In Situ Characterization of Twin Nucleation in Pure Ti Using 3D-XRD. *Metall Mater Trans A* 45A, 109-122.
- Kumar, M.A., Kanjarla, A.K., Niezgodá, S.R., Lebensohn, R.A., Tome, C.N., 2015. Numerical study of the stress state of a deformation twin in magnesium. *Acta Mater* 84, 349-358.
- Wang, L., Eisenlohr, P., Yang, Y., Bieler, T.R., Crimp, M.A., 2010. Nucleation of paired twins at grain boundaries in titanium. *Scripta Mater* 63, 827-830.

Wang, L.Y., Barabash, R., Bieler, T., Liu, W.J., Eisenlohr, P., 2013. Study of Twinning in alpha-Ti by EBSD and Laue Microdiffraction. Metall Mater Trans A 44a, 3664-3674.

REVIEWERS' COMMENTS:

Reviewer #1 (Remarks to the Author):

The reviewer would like to thank the authors for their response.

Average stress-strain curve could have been very much helpful as the applied stress is one of the key factors that all of the grains have in common; it also helps understand the level of stress concentration close to twins which can be used for model validation.

Given that the authors emphasize on calculating dislocation density as the novelty of this work in comparison to Balogh's work, it seems odd to use RSS values for this. The use of calculated lattice rotations with Nye tensor would have been another approach to have an estimation of dislocation densities as lattice rotations are normally 10x bigger than elastic strains and accuracy as well as precision in measuring them are much higher. This method is consistent with the one used in other diffraction based experimental techniques. However, I can understand why they used Eq. 1.

Reviewer #2 (Remarks to the Author):

The concerns raised by the reviewer have been addressed clearly in the revised manuscript and the response letter. The reviewer recommends the publication of the revised manuscript.

Minor revision:

line 263 on page 11, it should be "our study suggests that ...".

Response to Reviewer - 1

The reviewer would like to thank the authors for their response. Average stress-strain curve could have been very much helpful as the applied stress is one of the key factors that all of the grains have in common; it also helps understand the level of stress concentration close to twins which can be used for model validation.

Given that the authors emphasize on calculating dislocation density as the novelty of this work in comparison to Balogh's work, it seems odd to use RSS values for this. The use of calculated lattice rotations with Nye tensor would have been another approach to have an estimation of dislocation densities as lattice rotations are normally 10x bigger than elastic strains and accuracy as well as precision in measuring them are much higher. This method is consistent with the one used in other diffraction based experimental techniques. However, I can understand why they used Eq. 1.

Response: We thank the referee for his/her comments. As we mentioned in the previous revision, we do not perform the actual in-situ experiment and thus we cannot provide the in-situ stress-strain curve and so the applied stress.

The dislocation density can be, in principle, calculated either using diffraction line profile analysis or lattice rotations-based Nye tensor analysis or generalized Taylor law. The primary goal of this work is to map the local stresses in the vicinity of a twin in the bulk polycrystal. Thus, motivates us to employ Taylor law for dislocation density calculations. Note that the dislocation density calculation using the lattice rotations with Nye tensor is sensitive to measurement volume size. At the same time, the Taylor law approach is independent of the measurement volume size.

Response to Reviewer - 2

The concerns raised by the reviewer have been addressed clearly in the revised manuscript and the response letter. The reviewer recommends the publication of the revised manuscript.

Minor revision:

line 263 on page 11, it should be "our study suggests that ...".

Response: Thanks for pointing out. We have corrected it in the revision.